# A Novel Fast Exact Subproblem Solver for Stochastic Quasi-Newton Cubic Regularized Optimization

## Abstract

In this work we describe an Adaptive Regularization using Cubics (ARC) method for large-scale nonconvex unconstrained optimization using Limited memory Quasi-Newton (LQN) matrices. ARC methods are a relatively new family of second-order optimization strategies that utilize a cubic-regularization (CR) term in place of trust-regions or line-searches. Solving the CR subproblem exactly requires Newton's method, yet using properties of the internal structure of LQN matrices, we are able to find exact solutions to the CR subproblem in a matrix-free manner, providing very large speedups. Additionally, we expand upon previous ARC work and explicitly incorporate first-order updates into our algorithm. We provide empirical results for different LQN matrices and find our proposed method compares to or exceeds all tested optimizers with minimal tuning.

## 1 Introduction

Scalable second-order methods for training deep learning problems have shown great potential, yet ones that build on Hessian-vector products may be prohibitively expensive to use. In this paper, we focus on algorithms that require information similar to Stochastic Gradient Descent (SGD) Ruder (2016), namely, stochastic gradients calculated on mini-batches of data. Quasi-Newton (QN) methods are a natural higher-level alternative to first-order methods, in that they seek to model curvature information dynamically from past steps based on available gradient information. Thus, they can work out of the box in the same settings as SGD with little model-specific coding required.

However, this comes with possible instability of the step size. Controlling the step size can be done using line-searches along a given direction $s$ or using trust-regions to find the best $s$ for a given step size. A relatively recent alternative to the mentioned approaches is known as cubic regularization Nesterov and Polyak (2006); Cartis et al. (2011); Tripuraneni et al. (2018), which shows very promising results. In detail, we study the minimization problem of

$$\underset{s \in \mathbb{R}^n}{\text{minimize}}\, m_k(s) \stackrel{\text{def}}{=} f(x_k) + s^T g_k + \frac{1}{2} s^T B_k s + \frac{1}{3} \sigma_k ||s||^3, \tag{1}$$

for a given $x_k$, where $g_k \stackrel{\text{def}}{=} \nabla f(x_k)$, $B_k$ is a Hessian approximation, $\sigma_k$ an iteratively chosen adaptive regularization parameter, and $f(x_k)$ is the objective function to minimize evaluated at $x_k$. Equation 1 is also known as the CR subproblem. Cubic regularization shows promise because it can be shown that if $\nabla^2 f$ is Lipschitz continuous with constant $L$, then $f(x_k + s) \le m_k(s)$ whenever $\sigma_k \ge L$ and $B_k s = \nabla^2 f(x)s$ Nesterov and Polyak (2006). Thus if the Hessian approximation $B_k$ behaves like $\nabla^2 f(x)$ along the search direction $s$, the model function $m_k(s)$ becomes an upper bound on the objective $f(x + s)$. In such cases a line-search would not be needed as reduction in $m_k(s)$ translates directly into reduction in $f(x + s)$, removing the risk that the computational work performed minimizing $m_k(s)$ is wasted.

We propose an efficient exact solver to Equation 1 using Newton's method which is tractable in large-scale optimization problems under near-identical conditions to those in which SGD itself is commonly applied. As Newton's method corresponds to much of the computation overhead when solving Equation 1, a dense approach such as that described in Cartis et al. (2011) would be

prohibitive. However, by exploiting properties of LQN methods described in Erway and Marcia (2015) and Burdakov et al. (2017) (and further applied in other papers Chen et al. (2014); pei Lee et al.; Lee et al. (2022)), we can instead perform Newton's method in a reduced subspace such that the cost per Newton iteration is reduced from $\mathcal{O}(mn)$ to $\mathcal{O}(m)$, where $n$ is the problem dimension and $m$ is the history size, commonly chosen to be an integer between 5 and 20. The full-space solution to Equation 1 can then be recovered for a cost identical to that of classic LQN methods.

To the best of our knowledge, all previous attempts to use LQN methods in the context of the ARC framework have necessarily had to change the definition of $m_k(s)$ in order to find an approximate solution Andrei (2021); Liu et al. (2021); Ranganath et al. (2022). Remarkably, we present a mechanism for minimizing $m_k(s)$ using similar computational efforts to a single matrix inversion of a shifted LQN matrix, which itself is a lower bound of the complexity of traditional LQN approaches. Further, we show that by applying Newton's method in the reduced subspace, we can achieve speed improvements of more than 100x over a naive (LQN inversion-based) implementation. In the numerical results section we further show that this modification permits the application of LQN matrices with exact cubic regularization as a practical optimizer for large DNNs.

## 2 RELATED WORK

Second-order methods in machine learning are steadily growing more common Berahas et al. (2021); Brust et al. (2017); Chen et al. (2020); Goldfarb et al. (2020); Ma (2020); Ramamurthy and Duffy (2016); Yao et al. (2021). Limited-memory SR1 (LSR1) updates are studied in the context of ARC methodology using a "memory-less" variant in Andrei (2021). As we will describe in Section 3, many QN methods iteratively update $B_k$ matrices with pairs $(s_k, y_k)$ such that $B_k s_k = y_k$, where $y_k$ denotes the difference of the corresponding gradients of the objective functions. In Andrei (2021), $y_k$ is a difference of the gradients of $m_k(s)$. Liu et al. (2021) approximately minimize $m_k(s)$ by solving shifted systems of form $(B_k + \sigma \|s_{k-1}\| I)s_k = -g$, where the norm of the previous step is used as an estimate for the norm of $\|s_k\|$ to define the optimal shift. As described in Theorem 1 in Section 3, the optimal solution necessarily satisfies a condition of the form $(B_k + \sigma \|s_k\| I)s_k = -g$. Since the norm of $\|s_k\|$ may vary greatly between iterations, this solution is a noisy approximation. They further simplify the sub-problem using only the diagonals of $B_k + \sigma \|s_{k-1}\| I$ when generating $s_k$.

Ranganath et al. (2022) solve a modified version of the problem using a shape-changing norm as the cubic overestimation that provides an analytical solution to Equation 1. They transform $m_k(s)$ using similar strategies to those advocated in this paper. However, this norm definition is dependent on the matrix $B_k$ and thus makes the definition of the target Lipschitz constant, $L$, dependent as well. A nontrivial distinction in our approaches is that theirs requires a QR factorization of matrices of size $n \times m$. This may be prohibitive for deep learning problems, which may have billions of parameters. Bergou et al. (2017) explores a similar idea of making the norm dependant on the QN matrix. In Park et al. (2020), the ARC framework with stochastic gradients is used with a Hessian-based approach first advocated by Martens et al. (2010). In this case, $\nabla^2 f(x)$ is approximated within a Krylov-based subspace using Hessian-vector products with batched estimates of $\nabla^2 f(x)$. They then minimize $m_k(s)$ with this small-dimensional subspace.

An alternative to ARC methods is the use of trust-regions or line-searches. Though fundamentally different approaches, we can often borrow technology from the trust-region subproblem solver space to adapt to the ARC context. For example, Brust et al. (2017) outlines mechanisms for efficiently computing $(B_k + \lambda I)^{-1} g$ and implicit eigendecomposition of $B_k + \lambda I$ when solving the trust-region subproblem of minimizing $q_k(s) \stackrel{\text{def}}{=} f(x_k) + s^T g_k + \frac{1}{2} s^T B_k s$ while subject to $\|s\| \leq \delta$. Burdakov et al. (2017) significantly reduces the complexity and memory cost of such algebraic operations while solving the same problem. We thus adopt select operations developed therein when applicable, to adapt the method of Cartis et al. (2011) to the LQN context. Unlike the approach advocated in Burdakov et al. (2017), we avoid inversions of potentially ill-conditioned systems to improve the stability of the approach while simultaneously reducing computation overhead costs.

Note that we further extend Cartis et al. (2011) to the stochastic optimization setting. Thus we also share relation to past stochastic QN approaches. Erway et al. (2020) use the tools described in Brust et al. (2017) to create a stochastic trust-region solver using LSR1 updates. Schraudolph et al. (2007) generalizes BFGS and LBFGS to the online convex optimization setting. Mokhtari and Ribeiro (2014) studies BFGS applied to the stochastic convex case and develops a regularization scheme to prevent

the BFGS matrix from becoming singular. Sohl-Dickstein et al. (2014) explores domain-specific modifications to SGD and BFGS for sum-of-functions minimization, where the objective function is composed of the sum of multiple differential subfunctions. Byrd et al. (2016) considers not using simple gradient differencing for the BFGS update, but instead more carefully building $(s_k, y_k)$ pairs using Hessian-vector products; Berahas et al. (2021) also explores a similar idea of carefully choosing $s_k$ and $y_k$. Wang et al. (2016) tries to prevent ill-conditioning of $B_k$ for BFGS updates, similar to Mokhtari and Ribeiro (2014), but explicitly for the nonconvex case. Keskar and Berahas (2016) present an optimizer designed specifically for RNNs that builds on Byrd et al. (2016).

**Our Contributions.**

1. A fast $\mathcal{O}(mn)$ approach for exactly solving the cubic regularization problem for any limited memory quasi-Newton approximation that lends itself to an efficient eigendecomposition such as LBFGS and LSR1,

2. A hybrid first and second-order stochastic Quasi-Newton ARC framework that is competitive with current SOTA optimizers,

3. Convergence theory that proves convergence in the nonconvex case, and

4. Strong empirical results of this optimizer applied to real-life nonconvex problems.

## 3 ALGORITHM

In this section we describe the proposed algorithm. We will first provide a brief introduction to LQN matrices, then describe how to exactly and efficiently solve Equation 1 when $B_k$ is defined by an LQN matrix. We will demonstrate that the computational complexity of ARCLQN (detailed in Algorithm 3) is similar to that of classical LQN solvers. Later in this section we describe how to solve the nonlinear optimization problem (Algorithm 2) using this subproblem solver. Until Section 3.2, for simplicity, we will motivate the problem by largely considering full-batch gradient descent. However, the techniques being developed in this paper will largely be applied in the stochastic setting.

Popular Quasi-Newton updates such as BFGS, DFP, and SR1 are based on iteratively updating an initial matrix $B_0 = \gamma I$ with rank one or two corrections with pairs $(s_k, y_k)$ such that the property $B_k s_k = y_k$ is maintained each update (Nocedal and Wright, 2006). For example, the popular SR1 update formula is given by the recursive relation:

$$B_{k+1} \leftarrow B_k + \frac{(y_k - B_k s_k)(y_k - B_k s_k)^T}{s_k^T (y_k - B_k s_k)}, \tag{2}$$

where $y_k \overset{\text{def}}{=} g_k - g_{k-1}$ and $s_k \overset{\text{def}}{=} x_k - x_{k-1}$. To verify that the update is well-defined,

$$\|s_k^T (y_k - B_k s_k)\| > \epsilon \|s_k\| \|y_k - B_k s_k\| \tag{3}$$

is checked with a small number $\epsilon$. If condition 3 is not satisfied, $B_{k+1} \leftarrow B_k$. This helps ensure that $B_k$ remains bounded. While for much of this paper we will focus on the SR1 update, we stress that the exact subproblem solver proposed in this section will hold for all QN variants described in Erway and Marcia (2015). We discuss many of such variants later in Section 3.2.

Note that if $B_k$ is explicitly formed, the computational and memory costs are at least $\mathcal{O}(n^2)$; as such, for large-scale problems, limited-memory variants are popular. For such cases, only the most recent $m \ll n$ pairs of $(s, y)$ are stored in $n \times m$ matrices $S_k$ and $Y_k$, where $S_k \overset{\text{def}}{=} (s_{k-m+1}, \ldots, s_k)$ and $Y_k \overset{\text{def}}{=} (y_{k-m+1}, \ldots, y_k)$. In the limited memory case, $B_k$ is never explicitly formed, and operations using $B_k$ are performed using only $\gamma$, $S$, and $Y$ using $\mathcal{O}(mn)$ operations. How this is done specifically for the cubic-regularized case will become clearer later in this section. Before proceeding, we will next briefly describe the approach used by Cartis et al. (2011) for the case where $B_k$ is dense. Later we describe how to adapt their dense approach to the limited-memory case.

### 3.1 SOLVING THE CUBIC REGULARIZED SUB-PROBLEM

In this section we focus on efficiently finding a global solution to the cubic regularized subproblem given in Equation 1, restated here for convenience:

$$\underset{s\in\mathbb{R}^n}{\text{minimize}} \; m_k(s) \stackrel{\text{def}}{=} f(x_k) + s^T g_k + \frac{1}{2}s^T B_k s + \frac{1}{3}\sigma_k\|s\|^3. \tag{1}$$

We start by describing a Newton-based approach proven to be convergent in Cartis et al. (2011). Though their approach targets dense matrices $B_k$ where Cholesky factorizations are viable, we subsequently show in this section how to efficiently extend this approach to large-scale limited memory QN matrices. The Newton-based solver for Equation 1 is based on the following theorem:

**Theorem 1** ((Cartis et al., 2011))**.** *Let $B_k(\lambda) \stackrel{\text{def}}{=} B_k + \lambda I$, $\lambda_1$ denote the smallest eigenvalue of $B_k$, and $u_1$ its corresponding eigenvector. A step $s_k^*$ is a global minimizer of $m_k(s)$ if there exists a $\lambda^* \geq \max(0, -\lambda_1)$ such that:*

$$B_k(\lambda^*)s_k^* = -g_k, \tag{4}$$

$$\|s_k^*\| = \frac{\lambda^*}{\sigma_k}, \tag{5}$$

*implying $B_k(\lambda^*)$ is positive semidefinite. Further, only if $B_k$ is indefinite, $u_1^T g_k = 0$, and $\|(B_k + \lambda_1 I)^\dagger g_k\| \leq -\lambda_1/\sigma_k$, then $\lambda^* = -\lambda_1$.*

For simplicity we define $s(\lambda) \stackrel{\text{def}}{=} -(B + \lambda I)^{-1}g$, where the pseudo-inverse is used for the case where $\lambda = -\lambda_1$. We can then see that for the case where $\lambda^* \geq -\lambda_1$, $s_k^*$ is given by the solution to the following equation:

$$\phi_1(\lambda) \stackrel{\text{def}}{=} \frac{1}{\|s(\lambda)\|} - \frac{\sigma}{\lambda} = 0. \tag{6}$$

Note the authors of Cartis et al. (2011) show that when $B_k$ indefinite and $u_1^T g = 0$, the solution $s_k^*$ is given by $s_k^* = s(-\lambda_1) + \alpha u_1$ where $\alpha$ is a solution to the equation $-\lambda_1 = \sigma\|s(-\lambda_1) + \alpha u_1\|$. That is, whenever Equation 6 fails to have a solution (the "hard-case"), $s_k^*$ is obtained by adding a multiple of the direction of greatest negative curvature to the min-two norm solution to Equation 4 so that Equation 5 is satisfied. The authors of (Cartis et al., 2011) thus apply Newton's method to $\phi_1(\lambda)$ resulting in Algorithm 1. This corresponds to Algorithm (6.1) of Cartis et al. (2011).

---

**Algorithm 1** Newton's method to find $s^*$ and solve $\phi_1(\lambda) = 0$

---

**if** $B$ indefinite, $u_1^T g = 0$ **then**
    **if** $\|s(-\lambda_1)\| < \frac{\lambda}{\sigma}$ **then**
        Solve $-\lambda_1 = \sigma\|s(-\lambda_1) + \alpha u_1\|$ for $\alpha$
        $s^* \leftarrow s(-\lambda_1) + \alpha u_1$
    **else**
        $s^* \leftarrow s(-\lambda_1)$
    **end if**
**else**
    Let $\lambda > \max(0, -\lambda_1)$.
    **while** $\phi_1(\lambda) \neq 0$ **do**
        Solve for $s$:

$$(B + \lambda I)s = -g. \tag{7}$$

        Let $B + \lambda I = LL^T$.

$$Lw = s. \tag{8}$$

        Compute the Newton correction

$$\Delta\lambda^N \stackrel{\text{def}}{=} \frac{\lambda\big(\|s\| - \frac{\lambda}{\sigma}\big)}{\|s\| + \frac{\lambda}{\sigma}\big(\frac{\lambda\|w\|^2}{\|s\|^2}\big)} \tag{9}$$

        Let $\lambda \leftarrow \lambda + \Delta\lambda^N$.
    **end while**
    $s^* \leftarrow s(\lambda)$
**end if**

---

At first glance, Algorithm 1 may not look feasible, as Equation 8 requires the Cholesky matrix $L$, which is only cheaply obtained for small dense systems. Looking closer, we note that to execute Algorithm 1 one need not form $s$ and $w$, as only their corresponding norms are needed to compute $\Delta \lambda^N$. Relevantly, it has been demonstrated that matrices in the Quasi-Newton family have compact matrix representations of the form

$$B = \gamma I + \Psi M^{-1} \Psi^T, \tag{10}$$

further detailed in Byrd et al. (2016). For example, for LSR1 $\Psi = Y - \gamma S$ and $M = (E - \gamma S^T S)$ where $E$ is a symmetric approximation of the matrix $S^T Y$, whose lower triangular elements equal those of $S^T Y$ (Erway and Marcia, 2015). They further show that for matrices of this class, an $\mathcal{O}(mn)$ calculation may be used to implicitly form the spectral decomposition $B = U \Lambda U^T$, where $U$ is never formed but stored implicitly and $\Lambda$ satisfies

$$\Lambda = \begin{pmatrix} \gamma I & 0 \\ 0 & \hat{\Lambda} \end{pmatrix} \tag{11}$$

where $\hat{\Lambda} \in \mathbb{R}^{k \times k}$ is the diagonal matrix defined in Erway and Marcia (2015) to be $\text{diag}(\lambda_1, ..., \lambda_m)$. Thus we know that $B$ will have a cluster of eigenvalues equal to $\gamma$ of size $n - k$. We can exploit this property to further reduce the computational complexity of Algorithm 1. Note again that $\Delta \lambda^N$ in Equation 9 can be computed as long as $\|s\|$ and $\|w\|$ are known. Using the eigendecomposition of $B_k$, we get

$$\|s\|^2 = g^T U (\Lambda + \lambda I)^{-2} U^T g$$
$$\|w\|^2 = s^T L^{-T} L^{-1} s = s^T (B + \lambda I)^{-1} s = g^T U (\Lambda + \lambda I)^{-3} U^T g.$$

Note here that $\lambda$ denotes the parameter optimized in Algorithm 1 and *not* the diagonal values of $\Lambda$. If we then define $U$ block-wise we can define the components $\hat{g}_1$ and $\hat{g}_2$ as follows:

$$U = \begin{pmatrix} U_1 & U_2 \end{pmatrix} \Rightarrow U^T g = \begin{pmatrix} U_1^T g \\ U_2^T g \end{pmatrix} = \begin{pmatrix} \hat{g}_1 \\ \hat{g}_2 \end{pmatrix}. \tag{12}$$

Thus we can compute $\|s\|$ and $\|w\|$ in $\mathcal{O}(m)$ operations assuming $\hat{g}$ is stored, giving

$$\|s\|^2 = \frac{\|\hat{g}_1\|^2}{(\lambda + \gamma)^2} + \sum_{i=1}^{m} \frac{\hat{g}_2(i)^2}{(\hat{\lambda}_i + \lambda)^2} \tag{13}$$

$$\|w\|^2 = \frac{\|\hat{g}_1\|^2}{(\lambda + \gamma)^3} + \sum_{i=1}^{m} \frac{\hat{g}_2(i)^2}{(\hat{\lambda}_i + \lambda)^3} \tag{14}$$

Using this, the computation cost of Newton's method is reduced to an arguably inconsequential amount, assuming that $\|\hat{g}_1\|$ and $\hat{g}_2$ required by Equations 13 and 14 can be efficiently computed. We dub this optimization the "norm-trick". We additionally note that $Y$ and $S$ both change by only one column each iteration of Algorithm 3 (defined later). Thus with negligible overhead we can cheaply update the matrix $\Psi^T \Psi \in \mathbb{R}^{m \times m}$ each iteration by retaining previously computed values that are not dependent on the new $(s, y)$ pair. Making the following two assumptions, we can then show that Algorithm 1 can be solved with negligible overhead compared to classical LQN approaches.

**Assumption 1.** *The matrix $T = \Psi^T \Psi$ is stored and updated incrementally. That is, if $\Psi$ has one column replaced, then only one row and column of $T$ is updated.*

**Assumption 2.** *The vector $\bar{u} = \Psi^T g$ is computed once each iteration of Algorithm 3 and stored.*

We note that classic LQN methods at each iteration must update $B_k$ and then solve a system of the form $s = -(B_k + \lambda I)^{-1} g$ for some $\lambda \geq 0$. This creates an $\mathcal{O}(mn)$ computational lower bound that we aim to likewise achieve when generating an optimal step for Equation 1. In contrast to the approach described in Ranganath et al. (2022) that uses a QR factorization of $\Psi$, we use an analogous approach to that described in Burdakov et al. (2017) for trust-region methods to perform the majority of the required calculations on matrices of size $m \times m$ in place of $n \times m$. This saves significantly on both storage and computational overhead. Note that unlike Burdakov et al. (2017) we do not explicitly compute $M^{-1}$ as we have found this matrix can periodically become ill-conditioned.

Using the techniques detailed here (but proven in Section D of the appendix), we can form a very efficient solver for Equation 1 (using a modified version of Algorithm 1). Using the norm-trick, we

can avoid explicitly forming $s$ and $w$, reducing complexity of a Newton iterate from $\mathcal{O}(mn)$ to $\mathcal{O}(m)$. Using Equations 13 and 14 and Assumptions 1-2, we can thus solve $\lambda^* = \sigma\|s^*\|$ from Algorithm 1 in $\mathcal{O}(m^3)$ additional operations once $T$ and $\bar{u}$ are formed. Finally, once $\lambda^*$ is recovered, we can form $s^*$ in $\mathcal{O}(mn)$, a single inversion of a shifted system, the same complexity of classical LQN approaches. Full derivation and proof are available in Section D of the appendix.

## 3.2 SOLVING THE NONLINEAR OPTIMIZATION PROBLEM

In this section we focus on solving the problem

$$\min_{x \in \mathbb{R}^n} f(x) = \sum_{i=1}^{N} f_i(x), \tag{15}$$

where $f_i(x)$ is defined as the loss for the $i$-th datapoint, using the subproblem solver defined in Section 3.1. We follow the ARC framework as described in Cartis et al. (2011), stated here as Algorithm 2. A benefit of the algorithm defined in Algorithm 2 is that first-order convergence is proven if $B_k$ remains bounded and $f(x) \in \mathcal{C}^1(\mathbb{R}^n)$. Thus the condition that $B_k = \nabla^2 f(x)$ is greatly relaxed from its predecessors such as Nesterov and Polyak (2006).

---

**Algorithm 2** Adaptive Regularization using Cubics (ARC). Blue text in Equation 18 indicates our modification to default to an SGD-like step on failure.

---

Given $x_0, \sigma_0 > 0, \gamma_2, \gamma_1, \eta_2 > \eta_1 > 0, \alpha > 0$, for $k = 0, 1, \dots$, until convergence,

    1. Compute update $s_k^*$ such that:

$$m_k(s_k^*) \leq m_k(s_k^c) \tag{16}$$

       where the Cauchy point $s_k^c = -\upsilon_k^c g_k$ and $\upsilon_k^c = \underset{\upsilon \in \mathbb{R}_+}{\arg\min}\, m_k(-\upsilon g_k)$.

    2. Compute ratio between the estimated reduction and actual reduction

$$\rho_k \leftarrow \frac{f(x_k) - f(x_k + s_k^*)}{f(x_k) - m_k(s_k^*)} \tag{17}$$

    3. Update

$$x_{k+1} \leftarrow \begin{cases} x_k + s_k^* & \text{if } \rho_k \geq \eta_1 \\ x_k - \alpha g_k & \text{otherwise} \end{cases} \tag{18}$$

    4. Set

$$\sigma_{k+1} \text{ in } \begin{cases} (0, \sigma_k] & \text{if } \rho_k > \eta_2 \\ [\sigma_k, \gamma_1 \sigma_k] & \text{if } \eta_2 \geq \rho_k \geq \eta_1 \\ [\gamma_1 \sigma_k, \gamma_2 \sigma_k] & \text{otherwise} \end{cases} \tag{19}$$

---

In Algorithm 2, we first solve the CR subproblem (Equations 1,16; Algorithm 1) to find our step, $s_k^*$. We then determine if the step is accepted by examining if the ratio between the decrease in the objective ($f(x_k) - f(x_k + s_k^*)$) and the predicted decrease in objective ($f(x_k) - m_k(s_k^*)$) is large enough (Equations 17-18). Then, depending on $\rho_k, \eta_1$ and $\eta_2$, we adjust our regularization parameter $\sigma_k$: the 'better' the step is, the more we decrease $\sigma_{k+1}$, and the worse it is, the more we increase it (Equation 19). The amount of increase and decrease is governed by two hyperparameters, $\gamma_1$ and $\gamma_2$.

We note one important modification to the ARC framework: if we find that $\rho < \eta_1$, we take an SGD step instead of just setting $x_k \leftarrow x_{k-1}$ (Equation 18). While, empirically, rejected steps are not common, we find that reverting to SGD in case of failure can save time in cases where $B_k$ is ill-conditioned. One may note that we have no guarantees that $f(x_k) - f(x_k - \alpha g_k) > 0$, which may seem to contradict the ARC pattern detailed in Cartis et al. (2011) which only accepts steps which improve loss. However, Chen et al. (2018) proves that in a trust-region framework, if you accept all steps, $\rho_k$ need only be positive half of the time for almost sure convergence (Paquette and Scheinberg (2020) proves a similar result for first-order methods). It has also been shown that noisy SGD steps improve performance of final solution quality (Zhang et al., 2017; Zou et al., 2021). Implementation details regarding Algorithm 2 can be found in Section B of the appendix and Algorithm 3.

Joining the optimizations presented in Section 3.1 with the modifications in Section 3.2, we can form the full ARCLQN algorithm, explicitly described in Algorithm 3. It is worth noting that while much of the above discussion assumes our Hessian approximation $B_k$ is an LQN matrix with a compact representation, this is not required. Indeed, any Hessian approximation which lends itself to a fast eigendecomposition and inversion may be applied to this modified ARC framework, with the caveat that Algorithm 1 may be slower if the norm-trick cannot be used. We explore this potential extension more in Section 4.3, where we use the positive-definite Hessian approximation proposed in Ma (2020). We also provide theoretical analysis of the proposed framework in Section E of the appendix, where we prove that under moderate assumptions ARCLQN converges in the nonconvex case.

---

**Algorithm 3** ARCLQN, our proposed algorithm for solving Algorithm 2 under memory constraints.

---

**Require:** Given $x_0$ : initial parameter vector
**Require:** $0 < \eta_1 < \eta_2$ : hyperparameters to measure the level of success of a step
**Require:** $\mathcal{D}, q$ : dataset and minibatch-size, respectively.
**Require:** $\sigma_0$ : starting regularization parameter
**Require:** $\epsilon, \delta$ : tolerance parameters
**Require:** $f(x, b)$ : objective function with inputs parameters $x$ and minibatch $b$
**Require:** $\alpha_1, \alpha_2$ : learning rates
 1: Initialize $B_0 = I$.
 2: **for** $k = 1, 2, \ldots$ **do**
 3:     Let $b_k$ be a minibatch sampled randomly from $\mathcal{D}$ of size $q$
 4:     $g_k \leftarrow \nabla_x f(x_{k-1}, b_k)$
 5:     Calculate $\lambda_1$ of $B_{k-1}$
 6:     Let $\lambda \leftarrow \max(-\lambda_1, 0) + \epsilon$
 7:     Compute $s_k^*$ (using Algorithm 1)
 8:     Calculate $\rho$ (as in Equation 17)
 9:     **if** $\rho \geq \eta_1$ **then**
10:         $x_k \leftarrow x_{k-1} + \alpha_1 s_k^*$
11:         $y \leftarrow \nabla_x f(x_k, b_k) - g_k$
12:         Update $B_k$ using $B_{k-1}, \alpha_1 s_k^*, y$ if update and resulting $B_k$ are well-defined
13:         **if** $\rho \geq \eta_2$ **then**
14:             $\sigma_k \leftarrow \max(\frac{\sigma_{k-1}}{2}, \delta)$
15:         **end if**
16:     **else**
17:         $\sigma_k \leftarrow 2 \cdot \sigma_{k-1}$
18:         $x_k \leftarrow x_{k-1} - \alpha_2 g_k$
19:         $y \leftarrow \nabla_x f(x_k, b_k) - g_k$
20:         Update $B_k$ using $B_{k-1}, -\alpha_2 g_k, y$ if update and resulting $B_k$ are well-defined
21:     **end if**
22: **end for**

---

## 4   Numerical Results

### 4.1   Comparison to SR1

We start by benchmarking the optimized CR subproblem solver alone, without integration into the larger ARCLQN optimizer.[1] These results are summarized in Table 1. All timing information is reported as the average across 10 runs. We see that the dense SR1 solver fails to scale to more than 10,000 variables. We also see that the traditional LSR1 solver becomes computationally prohibitive for higher dimensions. For example, when $n = 10^8$, the positive-definite test case takes 274 seconds to converge for the inversion-based solver, whereas following steps outlined in Section 3.1, it is reduced to 2.33 seconds, a speedup of over 100x. Considering that the CR subproblem represents the bulk of the computation of any given optimization step, this performance improvement greatly increases the scalability of the algorithm. In the next section, we use the enhancements highlighted here to provide preliminary results using Algorithm 3.

---

[1]All experiments in this section were run on two Intel Xeon Gold 6150 processors.

**Table 1:** Timing information for solving the CR subproblem, Equation 1. A hyphen indicates that the test did not terminate within 300 seconds. SR1 corresponds a dense SR1 implementation. LSR1 corresponds to ARCLQN without the norm trick. For limited memory experiments, $m = 3$ was used. Cases are detailed in section 3. All other columns correspond to the problem dimension, and entries correspond to time (in seconds) required to find the global minimizer $s^*$ using CPU.

| | | Time (in seconds) to solve CR subproblem of given dimension | | | | | | |
|---|---|---|---|---|---|---|---|---|
| Method | Case | 1e2 | 1e3 | 1e4 | 1e5 | 1e6 | 1e7 | 1e8 |
| SR1 | Hard | 3.71e-3 | 2.79e-1 | 1.03e2 | - | - | - | - |
| LSR1 | Hard | 4.47e-4 | 8.05e-4 | 3.16e-3 | 5.72e-3 | 4.05e-2 | 8.75e-1 | 7.41e0 |
| ARCLQN | Hard | 4.31e-4 | 7.34e-4 | 1.68e-3 | 4.47e-3 | 2.66e-2 | 6.44e-1 | 5.64e0 |
| SR1 | Indefinite | 1.93e-3 | 8.53e-2 | 1.17e1 | - | - | - | - |
| LSR1 | Indefinite | 1.49e-3 | 3.54e-3 | 1.90e-2 | 4.40e-2 | 7.78e-1 | 8.55e0 | 8.14e1 |
| ARCLQN | Indefinite | 8.02e-4 | 1.69e-3 | 1.99e-3 | 2.79e-3 | 1.90e-2 | 2.59e-1 | 2.39e0 |
| SR1 | Positive Definite | 1.39e-3 | 9.65e-1 | 9.23e1 | - | - | - | - |
| LSR1 | Positive Definite | 7.85e-3 | 1.64e-2 | 9.09e-2 | 1.48e-1 | 2.12e0 | 3.06e1 | 2.74e2 |
| ARCLQN | Positive Definite | 3.64e-3 | 6.03e-3 | 6.45e-3 | 7.90e-3 | 2.68e-2 | 2.70e-1 | 2.33e0 |

## 4.2 AUTOENCODING

We also experiment with using ARCLQN as an optimizer for an autoencoder, as detailed in Goodfellow et al. (2016).[2] We use CIFAR-10 Krizhevsky (2009) as our dataset and compare against a number of recent optimizers Kingma and Ba (2015); Ruder (2016); Ma (2020); Yao et al. (2021). Hyperparameters are detailed in Section C of the appendix. For this experiment, as our Hessian approximation, we use an LSR1 matrix Ramamurthy and Duffy (2016). Results are summarized in Figure 1 and Table 2. All numbers reported are averaged across 10 runs. It is worth noting that while LBFGS converges more rapidly (by number of steps), it is over twice as slow as our approach (by wall clock time) and suffers from numerical stability issues: of the 10 runs performed, 2 failed due to NaN loss.

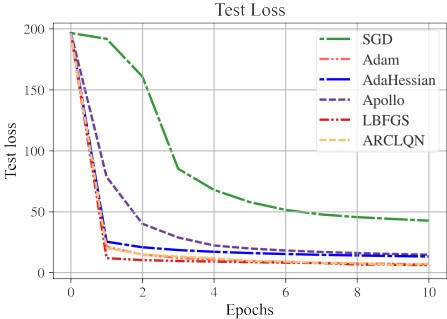

**Figure 1:** Test set loss of the trained CIFAR-10 autoencoder, evaluated at the end of each epoch.

**Table 2:** The amount of time taken to achieve best recorded test loss, relative to SGD. Timing information included all parts of training, but excluded calculation of test set loss.

| Optimizer | Cost ($\times$SGD) |
|---|---|
| SGD | 1.00 |
| Adam | 1.00 |
| AdaHessian | 1.01 |
| Apollo | 1.00 |
| LBFGS | 5.45 |
| ARCLQN | 1.95 |

## 4.3 IMAGE CLASSIFICATION

We additionally perform experiments on image classification using the ImageNet dataset Deng et al. (2009).[3] Motivated by Chrabaszcz et al. (2017) and a lack of computational resources, we resize ImageNet to 32x32. Following Ma (2020), we use a modified version of ResNet-18 (dubbed ResNet-110) adapted for smaller image sizes. Additionally, we use the best hyperparameters from Ma (2020), namely, the learning rate, epsilon, and momentum for all optimizers. Here we use the positive-definite diagonal Hessian approximation presented by Ma (2020) as our $B_k$. Unlike the other optimizers which received extensive hyperparameter searches, ARCLQN achieves strong results using the same

---

[2]All experiments in this section were run on a single NVIDIA V100 GPU over 2 days.

[3]All experiments in this section were run on a single NVIDIA A100 GPU over 3 days.

hyperparameters as in the CIFAR-10 experiments. This is of significance: our proposed method outperforms or compares to all optimizers considered without expensive hyperparameter tuning or hacking. It is worth noting that Apollo, the optimizer proposed by Ma (2020), requires a long warmup period for good performance. Our approach has no such requirement. We theorize this result is due to a combination of defaulting to SGD on failure and the ARC framework preventing any steps that would otherwise degrade performance. Finally, in Table 3, we can see that ARCLQN is associated with both the highest Top-1 accuracy and the lowest computational cost.

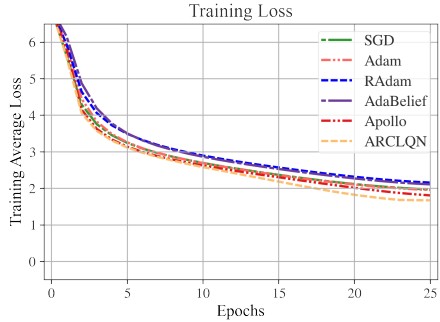
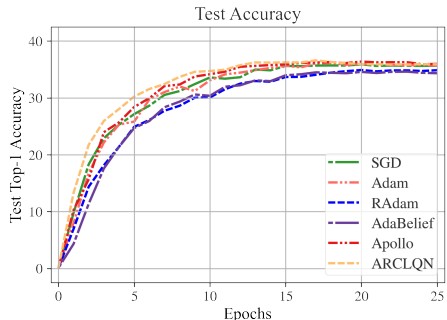

**Figure 2:** Average training loss for each epoch.   **Figure 3:** Test set accuracy for each epoch.

**Table 3:** Table summarizing performance of different optimizers. The cost corresponds to the amount of time taken to achieve best recorded test accuracy, relative to SGD. Timing information included all parts of training, but excluded calculation of test set accuracy.

| Optimizer | Best Top-1 Accuracy | Best Top-5 Accuracy | Cost (×SGD) |
|---|---|---|---|
| SGD | 35.89 | 60.09 | 1.0 |
| Adam | 36.12 | 59.85 | 0.86 |
| RAdam | 34.89 | **62.06** | 0.90 |
| AdaBelief | 34.61 | 61.38 | 0.96 |
| Apollo | 36.38 | 61.74 | 0.76 |
| ARCLQN | **36.59** | 61.76 | **0.69** |

## 5   CONCLUSION AND DISCUSSION

**Conclusion.**   We have introduced a new family of optimizers, referred to as ARCLQN, which utilize a novel fast large-scale solver for the CR subproblem. We demonstrate very large speedups over a baseline implementation, and we find that ARCLQN is competitive with modern first-order and second-order optimizers on real-world nonconvex problems with minimal tuning. To the best of our knowledge, ARCLQN is the first extension of the ARC framework to the limited memory case without major modification of the core framework. We additionally expand upon ARC, explicitly incorporating first-order updates into our methodology. Finally, we provide convergence analysis of the modified framework which proves convergence even for the nonconvex case.

**Limitations.**   While we have introduced an optimization framework that is applicable to any Hessian approximation that has a fast eigendecomposition, we do not consider Hessian approximations for which this information is not readily available. Additionally, if our Hessian approximation's eigenvalues differ greatly between steps, this can lead to oscillation in the calculated $\|s_k\|$. Future work may also include a wider variety of evaluated Hessian approximations, as there are many which were not tested here.

**Reproducibility Statement**   In this paper we provide all details to recreate our implementation, including algorithms matching how the optimizer is written in code. These are located in Section 3. We also provide all implementation details and hyperparameters in Section B and Section C respectively.

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

## A  ETHICAL CONSIDERATIONS

With the ever-increasing utilization and adoption of more powerful models, it is more and more important for authors to consider the ethical aspects of their work. Our work presented here is very general in nature, as it is possible to use ARQLQN as an optimizer for any function where gradient information is easily available. A major positive impact from this paper and subsequent research may be substantial reductions in power consumption, as is seen preliminarily in the significantly reduced runtime in Table 3. Additionally, as a general-purpose optimizer, this work may help progress societally beneficial research (such as in medicine). However, it also holds the ability to be misused (e.g., being used to train unethical models that discriminate based of protected personal attributes). This potential for misuse is inherent to all general-purpose optimization research. Avoidance of this may be impractical, and is beyond the scope of this work.

In our numerical results section (Section 4), we use two image based datasets: CIFAR-10 and ImageNet. Improper or careless use of datasets that contain sensitive information should be avoided where possible. We believe both CIFAR-10 and the version of ImageNet we used pose very little risk, as both of the datasets are at 32x32 resolution, hiding most sensitive information. The research done in this paper abides by the licenses provided by the authors of the datasets.

## B  IMPLEMENTATION DETAILS

Second-order methods can at times be unstable. To achieve good performance and stable training, it is important to use heuristics to prevent or alleviate this instability. For the sake of complete transparency, we share all used heuristics and modifications not explicitly detailed elsewhere. It is worth explicitly noting that none of the parameters below have been tuned for performance, and instead have been chosen either arbitrarily, or via test-runs on toy problems. There may be significant room for improvement with tuning of these parameters, and we leave that to future work.

### B.1  GENERAL DETAILS

An important detail is that we do not check if $\phi_1(\lambda) = 0$ when using Newton's method in Algorithm 1. Instead, we repeat the while loop until $\|s\| - \frac{\lambda}{\sigma} < \nu$. For CIFAR-10 and ImageNet experiments, we set $\nu = $ 1e-5. For comparison to SR1, we set $\nu = $ 1e-7. We empirically find that for larger scale problems, $\nu$ can be set higher, as $\phi_1(\lambda)$ does not change much at the final iterations of Algorithm 1. For CIFAR-10 experiments, we do not take an actual SGD step, but instead use an Adam step Kingma and Ba (2015). Finally, we also bound $\sigma$ above, as rarely $\rho < \eta_1$ can occur multiple times in a row, which can lead to many very small steps being taken with little effect on performance.

### B.2  LSR1 SPECIFIC

If we repeatedly take very similar or very small steps, we can run into issues with $S_k$ being singular or $B_k$ being ill conditioned. We use two heuristics to detect and fix this. First, before updating $B_k$ on lines 12 and 20 of Algorithm 3, we set $y \leftarrow \frac{y}{\max(\|s\|, \kappa)}$ and $s \leftarrow \frac{s}{\max(\|s\|, \kappa)}$. This prevents $B_k$ from becoming ill-conditioned if $\|s\|$ is very small. Additionally, we also reset $B_k$ if the minimum eigenvalue of $S_k^T S_k$ is less than $\kappa$. When we 'reset' $B_k$, we drop the first and last column of $S_k$ and $Y_k$ instead of setting $B_k \leftarrow I$; this helps us prevent resetting from destroying too much curvature information. We set $\kappa = $ 1e-7.

## C  HYPERPARAMETERS

This section will detail optimizer settings not otherwise explicitly mentioned in the paper. For all experiments, default dataset splits are used.

### C.1  CIFAR-10.

The hyperparameters used to generate Figure 1 can be found in Table 4. For CIFAR-10 experiments, $\alpha_2 = 0.005$ for ARCLQN was set arbitrarily. For ARCLQN, we use $\eta_1 = 0.05, \eta_2 = 0.6$; these

hyperparameters were not tuned, but instead set to be similar to Cartis et al. (2011). For all optimizers with momentum, $\beta$ values were left at their defaults and not tuned. Since the Apollo paper emphasizes that warmup is extremely important for their method, we use linear warmup over 500 steps Ma (2020). We performed a hyperparameter search over learning rates in $\alpha \in \{.001, .01, .1, .75, 1\}$ for all optimizers. For applicable optimizers, we also searched for the optimal $\epsilon \in \{1\text{e-4}, 1\text{e-8}, 1\text{e-16}\}$. For each optimizer, we chose the hyperparameter settings that lead to the lowest test loss after 10 epochs. For limited memory methods, we fixed the history size at $m = 5$. We use a convolutional neural network Goodfellow et al. (2016) with 3 convolutional layers, then 3 transposed convolutional layers. For all layers, padding and stride are set to 1 and 2 respectively. Between all layers (except the middle one), SeLU Klambauer et al. (2017) activations are used; the final layer uses a sigmoid activation. Layers have $3, 12, 24, 48, 24,$ and $12$ input channels respectively; minibatch size is fixed as 128. We use binary cross-entropy as our loss function.

**Table 4:** Hyperparameter settings for the optimizers used in the CIFAR-10 autoencoding experiments. A dash indicates that the optimizer does not have a given hyperparameter.

| Optimizer | $\alpha$ | $\beta$ | $\epsilon$ |
|---|---|---|---|
| SGD | 0.01 | 0.9 | - |
| Adam | 0.001 | $(0.9, 0.999)$ | 1e-8 |
| Apollo | 0.01 | 0.9 | 1e-8 |
| AdaHessian | 0.1 | $(0.9, 0.999)$ | 1e-4 |
| LBFGS | 1 | - | - |
| ARCLQN | $(.75, 0.005)$ | $(0.9, 0.999)$ | 1e-4 |

## C.2 IMAGENET

The hyperparameters used to generate Figure 2-3 and Table 3 can be found in Table 5. For AR-CLQN, we use the same hyperparameters as in Table 4. We use the cosine learning rate annealing scheduler Loshchilov and Hutter (2017), in line with Ma (2020), from which we take many of our hyperparameters.

**Table 5:** Hyperparameter settings for the optimizers used in the ImageNet classification experiments. A dash indicates that the optimizer does not have a given hyperparameter.

| Optimizer | $\alpha$ | $\beta$ | $\epsilon$ |
|---|---|---|---|
| SGD | 0.1 | 0.9 | - |
| Adam | 0.001 | $(0.9, 0.999)$ | 1e-8 |
| RAdam | 0.001 | $(0.9, 0.999)$ | 1e-8 |
| AdaBelief | 0.001 | $(0.9, 0.999)$ | 1e-4 |
| Apollo | .01 | 0.9 | 1e-4 |
| ARLQN | $(.75, 0.005)$ | - | 1e-4 |

## D PROOF OF SUBPROBLEM SOLVER COMPLEXITY

This section contains the full proof of many of the claims located in Section D. We will restate the assumptions made there, then progress with the proofs.

**Assumption 1.** *The matrix $T = \Psi^T \Psi$ is stored and updated incrementally. That is, if $\Psi$ has one column replaced, then only one row and column of $T$ is updated.*

**Assumption 2.** *The vector $\bar{u} = \Psi^T g$ is computed once each iteration of Algorithm 3 and stored.*

Unlike trust-region methods, the cubic-regularization search steps can sometimes quickly grow when negative curvature is found. To stabilize the approach we assume a safeguard is used (as in line-search methods Zhou et al. (2017)) that uniformly bounds the second-order correction matrix from singularity.

**Assumption 3.** *The final search direction has form*

$$s = -U^T \bar{\Lambda}^{-1} U g.$$

where $\bar{\Lambda}_{ii} = \max(\tau, \Lambda_{ii} + \lambda^*)$ *for small positive constant $\tau$ where $\Lambda$ is given by Equation 11 and $\lambda^*$ denotes the optimal shift found by Algorithm 1.*

The following theorem shows that given $T$ and $\bar{u}$ defined in Assumptions 1 and 2, that the optimal $\lambda^*$ can be cheaply obtained by Algorithm 1 using $\mathcal{O}(m^3)$ operations.

**Theorem 2.** *Suppose that $B \stackrel{\text{def}}{=} \gamma I + \Psi M^{-1} \Psi^T$ as defined in Equation 10 and that $(V, \Lambda)$ solves the generalized eigenvalue problem $Mv = \lambda Tv$. Then $U_2$ as defined in 12 is given by $U_2 = \Psi V$. Further the corresponding eigenvalues $\hat{\Lambda}$ from Equation 11 are given by $\hat{\Lambda} = (\gamma I + \Lambda^{-1})$. We can then recover $\hat{g}_2 = V^T \bar{u}$, and $\|\hat{g}_1\| = g^T g - \hat{g}_2^T \hat{g}_2$.*

*Proof.* Rather than inverting $M$ we can simply solve the generalized eigenvalue problem $[V, \Lambda] = \text{eig}(M, \Psi^T \Psi)$, where

$$
\begin{aligned}
MV &= \Psi^T \Psi V \Lambda \\
V^T M V &= \Lambda \\
V^T \Psi^T \Psi V &= I,
\end{aligned}
$$

where $\Lambda$ is the diagonal matrix of generalized eigenvalues for the system $Mv = \lambda \Psi^T \Psi v$. Then we have $U_2 = \Psi V$ implying

$$
BU_2 = \gamma U_2 + \Psi M^{-1} \Psi^T (\Psi V \Lambda \Lambda^{-1}) = \gamma U_2 + \Psi M^{-1} M V \Lambda^{-1} = U_2(\gamma I + \Lambda^{-1}).
$$

Thus we can can set $\hat{\Lambda}$ from equation 11 as $\hat{\Lambda} = (\gamma I + \Lambda^{-1})$. Further we can recover $\hat{g}_2 = U_2^T g = V^T \Psi^T g = V^T \bar{u}$ and $\|\hat{g}_1\| = g^T g - \hat{g}_2^T \hat{g}_2$. $\square$

Using the previous theorems and Equations 13 and 14 we can thus obtain $\lambda^* = \sigma \|s^*\|$ from Algorithm 1 in $\mathcal{O}(m^3)$ additional operations once $T$ and $\bar{u}$ are formed. We now show how to efficiently recover the optimal $s^*$ from Equation 1 using $\mathcal{O}(mn)$ operations.

**Theorem 3.** *Using the same assumptions and definitions as in Theorem 2, given any $\lambda > \max(0, -\lambda_1)$, the solution $s = -(B + \lambda I)^{-1} g$ is given by $-\dfrac{1}{\lambda + \gamma} g - \Psi V r$, where $r$ can be formed with $\mathcal{O}(m^2)$ computations.*

*Proof.* Note we can further save on computation by storing $\Psi^T g$ for later calculations when we recover the final search direction. Note that the very end we must form the search direction by solving the system $(B + \lambda I)s = -g$ with the optimal value of $\lambda$. This implies

$$
\begin{aligned}
s &= -U(\Lambda + \lambda I)^{-1} U^T g \\
&= -\frac{1}{\lambda + \gamma} U_1 U_1^T g - U_2 (\hat{\Lambda} + \lambda I)^{-1} U_2^T g \\
&= -\frac{1}{\lambda + \gamma} \left( U_1 U_1^T g + (U_2 U_2^T g - U_2 U_2^T g) \right) - U_2 (\hat{\Lambda} + \lambda I)^{-1} U_2^T g \\
&= -\frac{1}{\lambda + \gamma} g - U_2 [(\hat{\Lambda} + \lambda I)^{-1} - \frac{1}{\lambda + \gamma} I] U_2^T g \\
&= -\frac{1}{\lambda + \gamma} g - \Psi V \bar{E} V^T \Psi^T g,
\end{aligned}
$$

where $E = (\hat{\Lambda} + \lambda I)^{-1} - (\lambda + \gamma)^{-1} I$. $\square$

**Theorem 4.** *Let $(\lambda_1, u_1)$ denote the eigenpair corresponding to the most negative eigenvalue of the matrix $B$. Then, if $\gamma < \min(\text{diag}(\hat{\Lambda}))$, $u_1$ can be formed as $u_1 = \hat{r}/\|\hat{r}\|$ where $\hat{r} = (I - U_2 U_2^T) r$ for any vector $r$ in $\mathbb{R}^n$ such that $\|\hat{r}\| > 0$. Otherwise $u_1 = \Psi v_k$ where $v_k$ is a column of $V$ that corresponds to the smallest eigenvalue of $\hat{\Lambda}$.*

*Proof.* Note that $(I - U_2 U_2^T)$ is the projection matrix onto the subspace defined by the $\text{span}(U_1)$ implying $U_2 \hat{r} = 0$, then $B\hat{r} = \gamma U_1 U_1^T \hat{r} = \gamma(I - U_2 U_2^T)\hat{r} = \gamma \hat{r}$, since $\hat{r}$ has already been projected. Thus $\hat{r}$ is an eigenvector of $\gamma$. If $\gamma$ is not the smallest eigenvalue of $B$, then by design $u_1$ can be obtained from $U_2 e_1$ assuming the eigenvalues of $\hat{\Lambda}$ are sorted smallest to largest. $\square$

# E CONVERGENCE ANALYSIS

Several useful assumptions are given to establish the global convergence of Algorithm 3. Let $b$ below denote a minibatch.

**Assumption 4.** *The function $f(x)$ is bounded below by a scalar $L_f$.*

**Assumption 5.** *$\nabla f(x)$ is Lipschitz continuous for all $x$. That is, $\|\nabla f(x) - \nabla f(y)\| \leq L_1 \|x - y\|$.*

**Assumption 6.** *$H(x)$ is Lipschitz continuous for all $x$. That is, $\|H(x) - H(y)\| \leq L_2 \|x - y\|$.*

**Assumption 7.** *For any iteration k, we have that $E[g(x_k, b)] = \nabla f(x_k)$.*

**Assumption 8.** *For any iteration k, we have that the gradient for that minibatch $g(x_k, b)$ is bounded for all $x_k$. That is, $\|g(x_k, b)\| \leq L_g$.*

**Assumption 9.** *For any iteration k, we have that the hessian for that minibatch $H(x_k, b)$ is bounded for all $x_k$. That is, $\|H(x_k, b)\| \leq L_H$.*

From the results below, Algorithm 3 will iteratively reduce $f(x)$ with probability one when $\alpha_k$ goes to 0. Thus, it is expected that Assumptions 8 and 9 can be satisfied.

In Algorithm 3, $\alpha_1$ and $\alpha_2$ are used as the learning rates. To simplify the notation, $\alpha_k$ is used in this section as either of them for a given iteration $k$. The following assumption is then given.

**Assumption 10.** *The sequence of learning rates $\alpha_k$ in Algorithm 3 is chosen such that:*

1. $\sum_{i=1}^{+\infty} \alpha_i = +\infty$

2. $\sum_{i=1}^{+\infty} \alpha_i^2 < +\infty$

The first theorem below ensures Algorithm 1 will solve problem 1, and it can be found in Cartis et al. (2011).

**Theorem 5.** *(Cartis et al. (2011)) Algorithm 1 converges to the global solution of problem 1 whenever the initial $\lambda$ satisfies $\max(0, -\lambda_1) < \lambda < \sigma \|s\|$ where $\lambda_1$ denotes the smallest eigenvalue of $B_k$.*

Note that an initial $\lambda$ for the preceding theorem is easily found by choosing $\lambda$ suitably close to its lower bound.

The following lemma can be found in Berahas et al. (2021):

**Lemma 6.** *(Berahas et al. (2021)) Suppose that $x_k$ is generated by Algorithm 3 and assumption 5 holds, and also $B_k$ is the Hessian approximations updated by Equation 2 when the new curvature pair satisfies Equation 3. Then there exists a constant $c_1 > 0$ such that $\|B_k\| \leq c_1$.*

Using Lemma 6, the following important theorem can be derived.

**Theorem 7.** *Suppose that $x_k$ is generated by Algorithm 3 and assumption 5 holds, and also $B_k$ is the Hessian approximations generated by Equation 2 when the new curvature pair satisfies Equation 3, then there exists a constant $c_2 > 0$ such that $\|B_k + \sigma_k \|s_k\| I\| \leq c_2$*

*Proof.* We will first show that $\sigma_k \|s_k\|$ is always bounded for all $k$.

Let $B_k \overset{\text{def}}{=} U^T \Lambda U$, and where $\Lambda$ is a diagonal matrix and $U^T U = I$,

Then, we have:
$$B_k + \sigma_k \|x_k\| I = U^T (\Lambda + \sigma_k \|s_k\| I).U$$

Therefore,

$$(B_k + \sigma_k \|s_k\| I)(B_k + \sigma_k \|s_k\| I) = U^T (\Lambda + \sigma_k \|s_k\| I)(\Lambda + \sigma_k \|s_k\| I) U$$

Because of Lemma 6, there exists $\lambda_{min}^k$ and $\lambda_{max}^k$ such that

$$(\sigma_k \|s_k\| + \lambda_{min}^k)^2 \quad \leq \quad \frac{x^T (B_k + \sigma_k \|s_k\| I)(B_k + \sigma_k \|s_k\| I) x}{x^T x} \leq (\sigma_k \|s_k\| + \lambda_{max}^k)^2, \quad (20)$$

for all nonzero $x$. Note that $\lambda_{min}^k$ and $\lambda_{max}^k$ are bounded by $c_1$.

Set $x = s_{k+1}$ in Equation 20, and because of $s_{k+1}$ is solution of Problem 1, we have:

$$(\sigma_k\|s_k\| + \lambda_{min}^k)^2 \quad \leq \quad \frac{g_k^T g_k}{s_{k+1}^T s_{k+1}} \leq (\sigma_k\|s_k\| + \lambda_{max}^k)^2 \tag{21}$$

There are two scenarios now.

1. $\dfrac{g_k^T g_k}{s_{k+1}^T s_{k+1}}$ is bounded. Because of Equation 21, we can easily conclude that $\sigma_k\|s_k\|$ is bounded. Thus, the lemma follows.

2. $\dfrac{g_k^T g_k}{s_{k+1}^T s_{k+1}}$ is unbounded. That is, there exists $M_k \to \infty$ such that $\dfrac{g_k^T g_k}{s_{k+1}^T s_{k+1}} \geq M_k$.

   Because of Assumption 8, $g_k^T g_k$ is bounded. We have that $s_k^T s_k \to 0$ as $\dfrac{g_k^T g_k}{s_{k+1}^T s_{k+1}}$ is

   unbounded. Using Taylor expansion, we note that from Equation 17,

   $$\begin{aligned} \rho_k &= \frac{f(x_k) - f(x_k + s_k)}{f(x_k) - m_k(s_k)} \\ &= \frac{f(x_k) - m_k(s_k) - O(\|s\|^3)}{f(x_k) - m_k(s_k)} \end{aligned} \tag{22}$$

   But as $\|g_k\| \geq M_k\|s_k\|$, we have:

   $$\begin{aligned} f(x_k) - m_k(s_k) &= -g_k^T s_k - s_k^T B_k s_k \\ &\geq -M_k\|s_k\|^2 - c_1\|s_k\|^2, \end{aligned} \tag{23}$$

   when $\|s\|$ is small. Therefore, when $s_k$ is very small, combining Equation 22 and 23, we have:

   $$\begin{aligned} |\rho_k - 1| &= \frac{O(\|s\|^3)}{f(x_k) - m_k(s_k)} \\ &\leq \frac{O(\|s\|^3)}{M_k\|s_k\|^2 + c_1\|s_k\|^2} \end{aligned}$$

   Because $M_k \to \infty$, we can now conclude that $\rho_k \to 1$ as $\|s_k\| \to 0$. This means that $\sigma_k$ is bounded. Therefore when $\dfrac{g_k^T g_k}{s_{k+1}^T s_{k+1}}$ is unbounded, we have that $\|s_k\| \to 0$ and $\sigma_k$ is bounded. Therefore $\sigma_k\|s_k\|$ is bounded.

We can then conclude that $\|B_k + \sigma_k\|s_k\|I\|$ is bounded. $\qquad\square$

We now focus on the proof of the convergence of Algorithm 3.

**Lemma 8.** *Suppose that $x_k$ is generated by Algorithm 3 and assumptions 3, 5, 6, 7, 8, and 9 hold. Then, there exists $c_4 > 0$ such that:*

$$E[f(x_{k+1})] \quad \leq \quad E[f(x_k)] - \frac{\alpha_k}{c_3}E[\|\nabla f(x_k)\|^2] + \frac{\alpha_k^2}{2}c_4 \tag{24}$$

*Proof.* Because of Assumption 3, we have that there exists a $c_3 > 0$ such that

$$s^T(B_k + \sigma_k\|s_k\|I)s \geq c_3\|s\|^2 \tag{25}$$

holds for all the $s$ and iterations $k$.

Now suppose $x_{k+1} = x_k + \alpha_k s_k$. From the Taylor theorem, there exist $\theta_k$ such that:

$$f(x_{k+1}) \quad = \quad f(x_k) + \alpha_k s_k^T \nabla f(x_k) + \frac{\alpha_k^2}{2} s_k^T H(\theta_k)s_k. \tag{26}$$

Now there are two scenarios.

1. $\rho \geq \eta_1$, So $s_k$ is the solution of the cubic regularized subproblem. Thus, Equation 26 becomes

$$f(x_{k+1}) \quad = \quad f(x_k) - \alpha_k g_k^T (B_k + \lambda_k \|s_k\| I)^{-1} \nabla f(x_k) + \frac{\alpha_k^2}{2} s_k^T H(\theta_k) s_k. \quad (27)$$

Let $\nabla f(x_k) = g_k + \xi_k$. By Assumption 7, we have $E(\xi_k | x_k) = 0$.

So from Equation 27, now we have:

$$f(x_{k+1}) = f(x_k) - \alpha_k \nabla f(x_k)^T (B_k + \lambda_k \|s_k\| I)^{-1} \nabla f(x_k) +$$
$$\alpha_k \xi_k^T (B_k + \lambda_k \|s_k\| I)^{-1} \nabla f(x_k) + \frac{\alpha_k^2}{2} s_k^T H(\theta_k) s_k. \quad (28)$$

Note that because of Equation 25 and Assumption 8, we have:

$$c_3 \|s_k\|^2 \leq s_k^T (B_k + \lambda_k I) s_k \leq \|s_k\| \|g_k\| \leq L_g \|s_k\|. \quad (29)$$

Thus, $s_k$ is bounded.

Because of Equation 28, we further have:

$$f(x_{k+1}) \leq f(x_k) - \frac{\alpha_k}{c_3} \|\nabla f(x_k)\|^2 +$$
$$+\alpha_k \nabla f(x_k)^T (B_k + \lambda_k \|s_k\| I)^{-1} \xi_k + \frac{\alpha_k^2}{2} s_k^T H(\theta_k) s_k.$$

Because of Assumptions 6 and 9, the above equation becomes:

$$f(x_{k+1}) \leq f(x_k) - \frac{\alpha_k}{c_3} \|\nabla f(x_k)\|^2 +$$
$$\alpha_k \nabla f(x_k)^T (B_k + \lambda_k \|s_k\| I)^{-1} \xi_k + \frac{\alpha_k^2}{2} (L_2 \|s_k\| + L_H) \|s_k\|^2. \quad (30)$$

We now take the expected value of both sides of the above inequality, Because of $E(\xi_k | x_k) = 0$, we have:

$$E[f(x_{k+1}) | x_k] \leq f(x_k) - \frac{\alpha_k}{c_3} \|\nabla f(x_k)\|^2 + \frac{\alpha_k^2}{2} (L_2 \frac{L_g}{c_3} + L_H)(\frac{L_g}{c_3})^2. \quad (31)$$

2. $\rho < \eta_1$, that is, the SGD direction is used as $s_k$. Similarly, we have:

$$E[f(x_{k+1}) | x_k] \leq f(x_k) - \alpha_k \|\nabla f(x_k)\|^2 + \frac{\alpha_k^2}{2} L_g^2 \quad (32)$$

Thus, combining with Equations 31 and 32, we have that there exists $c_4 > 0$ such that:

$$E[f(x_{k+1}) | x_k] \quad \leq \quad f(x_k) - \frac{\alpha_k}{c_3} \|\nabla f(x_k)\|^2 + \frac{\alpha_k^2}{2} c_4. \quad (33)$$

Thus Lemma 8 holds. □

**Theorem 9.** *Suppose that $x_k$ is generated by Algorithm 3 and assumptions 3, 4, 5, 6, 7, 8, and 9 hold. Then, we have:*

$$\lim_{k \to \infty} E[\|\nabla f(x_k)\|] = 0. \quad (34)$$

*Proof.* Because of Lemma 8, we have:

$$\sum_{k=1}^{N} E[f(x_{k+1})] \quad \leq \quad \sum_{k=1}^{N} E[f(x_k)] - \sum_{k=1}^{N} \frac{\alpha_k}{c_3} E[\|\nabla f(x_k)\|^2] + \sum_{k=1}^{N} \frac{\alpha_k^2}{2} c_4.$$

That is,

$$\sum_{k=1}^{N} \frac{\alpha_k}{c_3} E[\|\nabla f(x_k)\|^2] \leq \sum_{k=1}^{N} E[f(x_k)] - \sum_{k=1}^{N} E[f(x_{k+1})] + \sum_{k=1}^{N} \frac{\alpha_k^2}{2} c_4.$$

Thus,

$$\sum_{k=1}^{N} \frac{\alpha_k}{c_3} E[\|\nabla f(x_k)\|^2] \leq f(x_1) - f(x_{N+1}) + \sum_{k=1}^{N} \frac{\alpha_k^2}{2} c_4. \tag{35}$$

Note:

$$E[\|\nabla f(x_t)\|^2] = \frac{1}{\left(\sum_{i=1}^{N} \alpha_i\right)} \sum_{k=1}^{N} \alpha_k E[\|\nabla f(x_k)\|^2]. \tag{36}$$

So, from Equation 35 and 36, we have:

$$
\begin{aligned}
\frac{\sum_{i=1}^{N} \alpha_i}{c_3} E[\|\nabla f(x_t)\|^2] &= \sum_{k=1}^{N} \frac{\alpha_k}{c_3} E[\|\nabla f(x_k)\|^2] \\
&\leq f(x_1) - f(x_{N+1}) + \sum_{k=1}^{N} \frac{\alpha_k^2}{2} c_4.
\end{aligned}
$$

That is,

$$E[\|\nabla f(x_t)\|^2] \leq \frac{c_3}{\sum_{i=1}^{N} \alpha_i}(f(x_1) - f(x_{N+1})) + \frac{\sum_{k=1}^{N} \alpha_k^2}{\sum_{i=1}^{N} \alpha_i} \frac{c_3 c_4}{2}. \tag{37}$$

Because of Assumption 4 and 10, we have:

$$\lim_{k \to \infty} E[\|\nabla f(x_k)\|^2] = 0. \tag{38}$$

Thus, the theorem follows. $\qquad\square$

