# OpenReview forum: "A Novel Fast Exact Subproblem Solver for Stochastic Quasi-Newton Cubic Regularized Optimization"
_ICLR.cc/2023/Conference — Submitted to ICLR 2023_

### Official Review · Reviewer_4R9H · 2022-10-23

**Confidence:** 3
**Correctness:** 2
**Technical Novelty And Significance:** 3
**Empirical Novelty And Significance:** 3
**Recommendation:** 5

**Clarity, Quality, Novelty And Reproducibility:**

Clarity: Many details need to be clarified. I list the ones I found as follows.

(1) (My major concern) It is still hard for readers to find out how to update $||\hat{g}_1||$ and $\hat{g}_2$ incrementally, and hard to see the whole procedure of the proposed exact cubic solver. I suggest to add an algorithm box either in main text or appendix to list all procedures of the proposed exact cubic solver, including the formulas for updating $||\hat{g}_1||$ and $\hat{g}_2$ incrementally.

(2) You could explain "ARCLQN" at its first appearance at the beginning of Section 3. For instance, "ARCLQN (Algorithm 3)".

(3) If convenient, right after eq. (11), you could write the definition of $\hat{\Lambda}$ or roughly how it is obtained.

(4) What are the sizes of $U_1$ and $U_2$ in eq. (12)? You could write them.

(5) Is $\hat{\Lambda}=\text{diag}(\hat{\lambda}_1,\ldots,\hat{\lambda}_m)$ correct? If yes, then you could mention that, and it seems that $k=m$ as $\hat{\Lambda}\in\mathbb{R}^{k\times k}$, yes?

(6) Does Algorithm 1 give the exact solution to the CR problem (1) within finitely many iterations? If yes, you could mention that and give iteration complexity.

(6) In step 4 of Algorithm 2, do we randomly select $\sigma_{k+1}$ in the correct interval?

(7) What equations are used to update $B_k$ in lines 12 and 20 of Algorithm 3?

(8) What does hard case mean in Table 1?

(9) What is the function $H$ in Assumptions 6 and 9 in Appendix E? Make sure every new terminologies and notations are defined upon their first appearance.

(10) In Theorem 2 in Appendix D, the generalized eigenvalue problem seems could be written as $MV=\Lambda TV$ to obtain $(V,\Lambda)$. Also, what is $\overline{u}$?

Quality: The claimed advantages of the proposed CR solver and ARCLQN algorithm look strong, but are not well supported due to the following reasons.

(1) (My another major concern) It seems that the authors claim Algorithm 1 to give exact CR solution, while Theorem 5 only converges to the exact solution. Could you explain that? Also non-asymptotic convergence rates or computation/sample complexity of Algorithms 1 and 3 are lacking, which have been obtained in previous cubic-regularization works. Therefore, the claimed higher efficiency of Algorithms 1 and 3 are not well supported unless faster convergence rate or computation/sample complexity can be proved.

(2) Assumption 8 implies that the objective function $f$ is Lipschitz continuous, which is rarely satisfied in many applications. Even quadratic function is not Lipschitz continuous.

(3) At the end of Section 2, the 4th item of "Our Contributions" could mention the advantage of the proposed ARCLQN demonstrated by the experiments.

Novelty: The ideas of matrix-free exact CR solver and incorporating SGD when the update of $B_k$ fails look novel.

Reproducibility: In the experiments, objective functions and some other hyperparameter choices such as $\sigma_0$, $\epsilon$, $\delta$, etc. could be provided to ensure reproducibility.

Minor comments:

(1) In Section 2, "Ranganath et al. (2022) solve a modified version", use "solves". There are other similar typos.

(2) Use $y_k, s_k$ instead of $y, s$ right after eq. (2).

(3) Change "remain" to remains and "paperd" to "paper" right after eq. (3).

(4) At the end of Theorem 1, should $g$ and $B$ be $g_k$ and $B_k$ respectively?

(5) In Algorithm 1, you could use "solve $-\lambda_1=\sigma\|s(-\lambda_1)+\alpha u_1\|$ to obtain $\alpha$" so that we know which variable to get. Similarly for eq. (7).

(6) Should $\alpha_1 s$ be $\alpha_1 s_k^*$ in line 12 of Algorithm 3? Also, "update and" seems could be removed.

(7) In the first line of Algorithm 2, $\gamma_2>\gamma_1>1$, yes?

**Strength And Weaknesses:**

Pros: The ideas of matrix-free exact CR solver and incorporating SGD when the update of $B_k~$fails look novel. Lit review looks very comprehensive. Experimental results look good.

Cons: There is lack of clarity, non-asymptotic convergence rate results and objective functions for experiments, as listed in "Clarity, Quality, Novelty And Reproducibility".

**Summary Of The Paper:**

The authors provide to their knowledge the first matrix-free Newton's method in reduced subspace to obtain exact solution of cubic-regularization (CR), which is much faster than the previous CR solvers. Then, the previous ARC algorithm is expanded to ARCLQN algorithm using LQN matrices, to incorporate SGD and the proposed exact cubic-regularization solver. ARCLQN is proved to converge to stationary point and empirically demonstrated more efficient than existing CR-based algorithms.

**Summary Of The Review:**

Based on "Strength And Weaknesses" above, I currently recommend borderline reject for this paper. If the authors can solve the above listed problems, especially my two major concerns (item 1 of clarity and quality), I would like to raise my rating.

---

> ### Author Response · Authors · 2022-11-19
> **Reviewer 4R9H Response**
>
> We thank the reviewer for their thoughtful comments and critiques. We have enumerated the concerns raised about the paper and responded to them below.
>
> (1) It is still hard for readers…
>
> We see your concern. Some specific details are kept abstract, such as the exact calculation of ||g_1|| and g_2. This is because the calculation of these is usually specific to the exact limited-memory Hessian approximation being used. However, ARCLQN does not depend on any specific Hessian approximation, and instead works wherever this information is available. We demonstrate this by using two different Hessian approximations in the paper, both LBFGS and the B_k proposed by (Ma 2020). The exact calculation of g_1 and g_2 is perhaps beyond the scope of this paper.
>
> (2) You could explain "ARCLQN" at its first appearance at the beginning of Section 3. For instance, "ARCLQN (Algorithm 3)".
>
> We agree and have made this edit in the paper.
>
> (3) If convenient, right after eq. (11)...
>
> Like the calculation of g_1 and g_2 before, the methodology to compute Lambda is dependent on the choice of B_k. Therefore, the ARCLQN algorithm itself is independent of how lambda is calculated: any Quasi-Newton matrix with a low-rank form is applicable. More information is available in Section at the beginning of Section 3, or near Equation 10.
>
> (4) What are the sizes…
>
> U_1 is of size (n, n - k). U_2 is of size (n, k).
>
> (5) Is $\hat \Lambda = $...
>
> Yes, that is correct. We have clarified this in the paper.
>
> (6) Does Algorithm 1…
>
> Algorithm (1) will not converge in general in finitely many steps. Empirically, we find it satisfies the equation to the tolerances of numerical precision within few iterations; Table 1 was generated with a tolerance (detailed in section B.1 of appendix) of 1e-7; lower tolerances are possible but may suffer from numerical precision issues.
>
> (6) In step 4 of Algorithm 2…
>
> No. The exact behavior is detailed in lines 14 and 17 of Algorithm 3.
>
> (7) What equations are used…
>
> This was purposefully left abstract; as was stated, any B_k which lends itself to an efficient eigendecomposition can be used. As such, this B_k should be updated in whatever way is appropriate for that Hessian. In CIFAR10 experiments, we use LBFGS. In ImageNet, we use the B_k presented in (Ma 2020).
>
> (8) What does hard case mean in Table 1?
>
> The “hard case” in Table 1 refers to the case where Equation (6) has no solution; this is described just below Equation (6). This has been clarified in the updated paper.
>
> (9) What is the function…
>
> The notation H(x) and g(x, b) are used for the Hessian of f(x), and the gradient of f(x) evaluated at a batch set b, respectively. We agree with the suggested change and this is reflected in the updated paper.
>
> (10) In Theorem 2 in Appendix D…
>
> $\bar u$ is defined in Assumption 2.
>
> (1) It seems that the authors claim Algorithm 1 to…
>
> Algorithm 1 involves the Newton method and it will iteratively find the approximated solution lambda* and s*. Theorem 5 gives its convergence result. When we say that we exactly solve the cubic regularization problem, we mean the ability to quickly find the desired solution to .  We will use the norm trick to update lambda. That is, we use Equations 13  and 14 to find the norm of s and w. We then update lambda by using Equation (9). Only when lambda* is found, Equation (7) and (8) is used to find s*.
> In terms of higher efficiency of Algorithms 1 and 3, as explained at Page 5 and Page 6 in paper, we use the norm-trick to reduce the complexity of a Newton iteration from O(mn) to O(m). We explicitly solve Equation (8) for s* only when the optimal lambda* is found. We use Equations 13  and 14 to find the norm of s and w. We then update lambda. Thus, the computation cost of Newton’s method is reduced to an arguably inconsequential amount. Also in contrast to the approach described in Ranganath et al. (2022), we use an analogous approach to that described in Burdakov et al. (2017) for trust-region methods to perform the majority of the required calculations on matrices of size m × m in place of n × m. This saves significantly on both storage and computational overhead. Unlike Burdakov et al. (2017) we do not explicitly compute M^{−1} as we have found this matrix can periodically become ill-conditioned.
>
> (2) Assumption 8 implies that the objective function is Lipschitz continuous…
>
> For a minimization algorithm, over the iterations f(x) is monotonically decreased and its gradient approaches 0. Thus, we assume that g(xk, b) is bounded for all xk and batch b, which can be satisfied in practice.
>
> Reproducibility: In the experiments, …
>
> We agree with your comment. In Sections B-C, we tried to detail every single variable not otherwise explicitly defined in the paper. We forgot to include s_0, which has now been added.
>
> – Minor comments:
>
> We have considered your comments and made changes where suitable.
>
> If there are any remaining concerns or questions, we are more than happy to continue discussing!

---

### Official Review · Reviewer_VDMs · 2022-10-25

**Confidence:** 3
**Correctness:** 2
**Technical Novelty And Significance:** 2
**Empirical Novelty And Significance:** 2
**Recommendation:** 3

**Clarity, Quality, Novelty And Reproducibility:**

The novelty of this work is limited since it provides nothing new but efficient implementation of an existing algorithm.

**Strength And Weaknesses:**

Pros:

- The algorithm design is clear.

Cons:

- This work may lack importance. Although making an efficient cubic-subproblem solver is a very important task in second-order optimization research, the main contribution of this work seems only to be providing an efficient implementation of solving a Newton method by adapting the low-rank structure, which does not contribute a lot of new knowledge to the optimization community.

- The presentation of this paper is very confusing. For instance,

-- Page 5, the authors keep mentioning matrices $S$ and $Y$. What are their definitions?

-- Page 5, the matrix $\Phi$ is defined in (10), which serves as one component from the decomposition of $B$. However, later in Assumption 1, the authors mention 'replace columns in $\Phi$), which confuses me since $\Phi$ should be some fixed matrix.

- I suggest the authors compare their L-BFGS algorithms with some Krylov subspace-based algorithms for solving the cubic subproblems, such as [1]. It is interesting to see how their computational complexity differs from each other, given the fact that the approximate Krylov subspace-based algorithms actually perform very well in practice.

[1] Kohler, Jonas Moritz, and Aurelien Lucchi. "Sub-sampled cubic regularization for non-convex optimization." International Conference on Machine Learning. PMLR, 2017.

- The experiments need more demonstration. For instance,

-- In Table 3, I do not understand why SGD achieves the highest cost, compared with other algorithms. Are they supposed to run the same 25 epochs? If so, it is strange because SGD should enjoy the lowest cost since any other algorithms need additional computation besides the computation of gradients.




**Summary Of The Paper:**

This work studies using L-BFGS method to solve the cubic subproblem in the second-order optimization. The authors provide theoretical analysis to suggest their method converges. Experiment results show their algorithm is empirically better than existing approaches.

**Summary Of The Review:**

This paper studies how to efficiently solve the cubic sub-problem given the limited-memory limit. However, the contribution of this work is somehow limited, and its presentation also needs to be further improved. Thus I recommend a reject to this paper.

---

> ### Author Response · Authors · 2022-11-19
> **Reviewer VDMs Response**
>
> We thank the reviewer for their thoughtful comments and critiques. We have enumerated the concerns raised about the paper and responded to them below.
>
> – This work may lack importance…
>
> The optimization of existing algorithms is an important problem. As seen in Table 1, in the indefinite Hessian case, our optimized algorithm is more than 100x faster than the unoptimized one in the large-scale test with vector dimension 1e8. Beyond just an observable speedup, the proposed algorithm avoids calculation of information previously “required”: by using the norm-trick, we avoid ever needing to calculate s or w to complete an iteration of Algorithm 1. Additionally, we expanded upon ARC by incorporating an SGD-like backup step which reduces the amount of wasted computation, something especially important in AI. Developing this algorithm and subsequently optimizing it allows application of limited-memory quasi-Newton matrices with ARC to DNNs. We demonstrate the value of this innovation by achieving both the fastest convergence and best top-1 accuracy performance in the ImageNet experiments. Finally, as was discussed in Section 2 of the paper, many other papers exist which solve a modified or easier version of this problem, motivating the fact that a solution to such a problem is impactful.
>
> -- Page 5, the authors keep mentioning matrices…
>
> $S_k$ and $Y_k$ are defined at the bottom of page 3. Often, when the iteration is obvious, we drop the $k$ for simplicity. We have clarified this in the text.
>
> -- Page 5, the matrix is defined in (10)...
>
> For a given iteration, Psi is a fixed matrix. When we move to the next iteration, one column of Psi will be replaced because of the limited memory Quasi Newton update which only considers the most recent m updates.
>
> -- In Table 3, I do not understand why SGD achieves the highest cost…
>
> As was stated in the caption of Table 3, the “cost” column corresponds to the amount of time taken to achieve best recorded test set accuracy, not total wall-clock time, for 25 epochs, relative to how long this took SGD. For example, Adam has a cost of 0.86 because it took Adam 0.86 times the amount of seconds that it took SGD to reach the lowest test loss by wall-clock time. SGD converged after 25 epochs, which is why the chart stops there.
>
> If there are any remaining concerns or questions, we are more than happy to continue discussing. Thank you for the comments and thoughts.

---

### Official Review · Reviewer_ZmCM · 2022-10-25

**Confidence:** 3
**Correctness:** 3
**Technical Novelty And Significance:** 2
**Empirical Novelty And Significance:** 2
**Recommendation:** 6

**Clarity, Quality, Novelty And Reproducibility:**

It is easy to read to paper.
The idea is new.

**Strength And Weaknesses:**

The solver proposed in this paper can much reduce the computation cost and is easy to implement.
I believe this method has potential applications in non-convex optimization.

**Summary Of The Paper:**

This paper proposes a novel fast exact subproblem solver for stochastic Quasi-Newton cubic regularized optimization.
The solver can much reduce the computation cost and is easy to implement.
I believe this method has potential applications in non-convex optimization.

**Summary Of The Review:**

The solver proposed in this paper can much reduce the computation cost and is easy to implement.
I believe this method has potential applications in non-convex optimization.

---

> ### Author Response · Authors · 2022-11-19
> **Rebuttal for ZmCM**
>
> We thank the reviewer for their comments. If you have any further thoughts about the paper, we would love to discuss them with you, as no specific qualms or problems with the paper were presented.

---

### Decision · Program_Chairs · 2023-01-20

**Decision:**

Reject

**Justification For Why Not Higher Score:**

N/A

**Justification For Why Not Lower Score:**

N/A

**Metareview: Summary, Strengths And Weaknesses:**

This work mainly proposes a method to solve the subproblems of the cubic regularized Quasi-Newton method, where L-BFGS method is used to approximate Hessian matrix.  A few related asymptotic convergence results are provided to support the method.

From the reviewers, the main complaint is that the novelty of this paper does not meet the standard of ICLR. That is, the paper only focuses on a very specific aspect of the CRN method. The paper describes a way to make the CRN subproblem with L-BFGS style approximate Hessian matrix. More specifically, using L-BFGS style Hessian approximation is also known, the authors only propose a way to make update for the subproblem.

Another issue is that, due to the use of inexact Hessian matrix, the $const\cdot\|x_k-x_{k+1}\|^3$ sufficient descent of the original CRN method is missing. In order to guarantee the convergence, an EXACT function evaluation is needed to certify descent, see Eqn. (17). This is OK for deterministic optimization. However, this is fatal for the stochastic case. Evaluating the function and certifying a sufficient descent exactly is very hard for stochastic optimization. When we look at the neural network ERM example considered in this paper, making a function evaluation (forward pass) is already half the cost of the full gradient evaluation (backward pass). This makes the stochastic sampling meaningless.

Moreover, the theoretical contribution is also weak, only asymptotic convergence is obtained.


**Summary Of Ac-Reviewer Meeting:**

N/A